

Foliar nutrient uptake from dust sustains plant nutrition
Anton Lokshin[1,2*], Daniel Palchan[2], Elnatan Golan[3], Ran Erel[3], Daniele Andronico[4], and Avner Gross[1]
1.   The Department of Environment, Geoinformatics and Urban planning Sciences, Ben Gurion University of the
4         Negev; Beer Sheva, Israel.
2.   The Department of Civil Engineering, Ariel University; Ariel, Israel.
3.   Institute of Soil, Water and Environmental Sciences, Gilat Research Center, Agricultural Research Organization;
7         Gilat, Israel.
4.   Istituto Nazionale di Geofisica e Vulcanologia, Sezione di Catania-Osservatorio Etneo, Rome, Italy.
**\* Corresponding author:** Lokshinanton@gmail.com










Abstract

Mineral nutrient uptake from soil through the roots is considered the exclusive nutrition pathway for vascular
terrestrial plants. Recently, desert dust was discovered as an alternative nutrient source to plants, through direct
uptake from dust deposited on their foliage. Here we studied the uptake of nutrients from freshly deposited desert
and volcanic dusts by chickpea plants under ambient and future elevated levels of atmospheric $CO_2$, through the
roots and directly through the foliage. We found that within weeks, chickpea plants acquired phosphorus (P) from
dust only through foliar uptake under ambient conditions, and P, Iron (Fe) and Nickel (Ni) under elevated CO2
conditions, significantly increasing their growth. Using additional chickpea variety with contrasting leaf properties
we have shown that the foliar nutrient uptake pathway from dust is facilitated by leaf surface chemical and
physiological traits such as low pH and trichome densities. We analyzed Nd radiogenic isotopes extracted from
plant tissues after dust application to assess the contribution of mineral nutrients that were acquired through the
foliage. Our results suggest that foliar mineral nutrient uptake from dust is an important pathway, that may play
an even bigger role in an elevated $CO_2$ world.
**Keywords:** plant nutrition; Nd isotopes; hidden hunger; foliage; elevated $CO_2$





**Introduction**
Vascular plants obtain carbon (C) from the atmosphere and most of their mineral nutrients from the soil. Hence,
it is generally thought that mineral nutrients such as phosphorus (P), potassium (K), iron (Fe), and other macro
and micronutrients are acquired predominantly through the plant's roots system (Marschner et al., 1997). Evidence
gathered in recent decades demonstrates that the atmosphere is an important source for mineral-nutrients to
terrestrial ecosystems via dust deposition (Chadwick et al., 1999; Goll et al., 2023; Gross et al., 2015; Van
Langenhove et al., 2020; Okin et al., 2004; Palchan et al., 2018). The concentration of P (and other nutrients) in
mineral atmospheric particles such dust and volcanic ash are enriched relative to most soils and are important
plant nutrient source, especially when soil fertility is low or in dusty regions (Arvin et al., 2017; Bauters et al.,
2021; Ciriminna et al., 2022; Eger et al., 2013; Gross et al., 2016b). In a montane environment in California, dust
P contribution to plants was documented to outpace the contribution from weathering of host bedrock (Arvin et
al., 2017). In a recent study we discovered that certain crop plants can gain P directly from the atmospheric dust,
via particles that accumulate on their leaves (Gross et al., 2021a; Lokshin et al., 2024b). Over short time scales,
foliar uptake was found as the only P uptake pathway from biomass fire ash particles (while the roots played a
negligible role (Lokshin et al., 2024a, b). These recent findings highlights the need to better understand the role
of the contribution of nutrient uptake from dust through the foliage (i.e., direct foliar nutrient uptake), a process
that has been traditionally overlooked and has never been quantified before, even though foliar fertilization has
been a well-known agricultural practice for many decades (Fageria et al., 2009; Ishfaq et al., 2022; Bukovac &
Wittwer, 1957; Wittwer & Teubner, 1959). In the context of climate change, the foliar pathway may be even more
pronounced for plants that will grow under elevated $CO_2$ (e$CO_2$) conditions because of two documented
phenomena: the 'dilution' effect, where accumulation of C exceeds that of mineral nutrients (Loladze, 2014), and
partial inhibition of key root uptake mechanisms (Gojon et al., 2023), together with soil fertility degradation (Lal,
2009; St.Clair and Lynch, 2010). These changes will drive plants to adapt and search for other nutrient uptake
pathways. The use of the foliar pathway under e$CO_2$ may offset the alarming phenomenon where an increasing
production of carbohydrates causes dilutes the concentration ofmacro and micronutrients such as P, Fe, calcium
(Ca), magnesium (Mg), K, zinc (Zn), copper (Cu), nickel (Ni) and others that are vital for the floral ecological
systems (Clarkson and Hanson, 1980) and for their dependent human and livestock nutrition (Lal, 2009; Loladze,
2002; Lowe, 2021).In this experiment, we cultivated C3 chickpea plants (specifically the 'Zehavit' variety, a
widely grown modern cultivar) under both current atmospheric $CO_2$ concentrations and elevated $CO_2$ conditions
in a controlled glasshouse environment. The primary objective was to demonstrate, describe and quantify nutrient
uptake via the leaves. We introduced two distinct types of mineral dust to the plants, applying them either to the
surface near the root zone or directly onto the leaves. The two dust types were representative of major atmospheric
particulate matter sources, namely desert-derived dust and volcanic ash (referred to as "dust" hereafter), with
average annual global emissions estimated at 3,000 Tgy$^{-1}$ and 300 Tgy$^{-1}$, respectively (Kok et al., 2021;
Langmann, 2013a).
We studied leaf traits that facilitate the foliar nutrient uptake from dust, its impact on plants' ionome (i.e., plant
elemental status), and used Nd radiogenic isotopes, present within the dust particles and characterized by distinct
isotopic values, to quantify the contribution of the foliar pathway. In addition, we used a non-responsive genotype,
'CR934', of the wild progenitor C. reticulatum, to study the impact of dust deposition on plant nutrition and



compare leaf properties under dust foliar fertilization between the modern chickpea cultivar and its wild
counterpart.



**Materials and Methods**
**Experimental design**
To study the impact of dust deposition on plant nutrition, two chickpea genotypes (*Cicer*) from the Hebrew
University of Jerusalem chickpea collection were selected based on preliminary experiments, showing contrasting
response to foliar dust application (Gross et al., 2021b). The non-responsive genotype: 'CR934', of the wild
progenitor *C. reticulatum* accession, sampled near Savur, Turkey. And the responsive genotype 'Zehavit' that is
a modern, high yield line, and considered popular among the Israeli growers. To test the biogeochemical response
of the foliar nutrient uptake we used the 'Zehavit' genotype. Experiments were conducted at Gilat Research Center
in southern Israel (31°21' N, 34°42' E) in two separate glasshouse rooms. In one room we set the $CO_2$
concentration to the ambient 412 ppm ($aCO_2$) and in the other room to elevated 850 ppm ($eCO_2$), simulating
current and future earth $CO_2$ concentrations based on high emissions scenario (business as usual, SSP 8.5, IPCC,
2021). Following germination, plants were cultivated in 72 pots containing inert media (perlite 206, particle size
of 0.075–1.5 mm; Agrekal, HaBonim, Israel). The pot size was 3 litter, with sufficient room for root growth during
the experimental period. The description of the growing conditions and fertigation nutrient supplement is provided
in Lokshin et al. (2024a).
At 14 days after germination, when plants were early in the vegetative phase (two or three developed leaves), we
changed the nutrient solution of 60 out of the 72 pots to P deficient fertigation (P concentration of 0.1 mg $L^{-1}$) to
create P starvation (-P treatment). Preliminary tests showed that our -P deficient media allows chickpea plants to
continue their growth cycle and increase their responsiveness for dust application and $eCO_2$ condition (Gross et
al., 2021, Lokshin et al., 2024). The remaining 12 pots continued to receive the full P sufficient nutrient media
(+P treatment). Plants fertigated with -P solution started to show P-deficiency symptoms such as chlorosis of
mature leaves, slight symptoms of necrotic leaf tips and an overall decrease in biomass accumulation at 35 days
after germination. At this stage we applied desert dust and volcanic ash on the -P plants.
Of a total number of plants (72) 48 were treated with dust and 24 served as untreated control group. Twenty-four
plants were applied with dust on their foliage by manually sprinkling dust through a 63$\mu$m sieve in proximity to
the foliage and 24 plants received root treatment by applying dust through a 63 $\mu$m sieve on the surface of the pot,
followed by gentle mixing of the surface to sink the dust particles deeper to enhance the physical contact between
the roots and the particles, thereby increasing the chances of having a more significant impact. Among the control
plants, 12 plants received the +P fertigation and 12 additional plants received -P fertigation. Each treatment group
was divided into two $CO_2$ levels, 36 plants in each $CO_2$ growing room. The plants were harvested 10 days after
the last dust application (55 days after germination). To ensure that nutrients from dust particles were not washed
by the irrigation during the experiment, we monitored the total P (i.e., P that dissolves in strong acid) in the water
that drained from the pots (Longo et al., 2014; Gross et al., 2015) throughout the experiment.

`



We performed a parallel experiment under $aCO_2$ where we grew six additional plants, in larger 5 L pots, filled
with soil, to test whether our findings also apply to natural soil conditions (Fig. S1).
**Mineral dust material**
We applied plant foliage and the area near plants' roots, with desert dust and volcanic ash, the two main mineral
dust types in the atmosphere (Langmann, 2013b). To achieve enough mass for our experiment, we produced dust
analogs from surface desert soil and surface volcanic ash soil, following common procedures described by others
(Gross et al., 2021b; Stockdale et al., 2016). The desert dust analog surface soil was collected from the southern
Israel Negev desert (30°320N 34°550E) (Gross et al., 2021b). Chemical and mineralogical properties of the
resulted dust are comparable to dust collected in the Sahara and other places in the Middle East (Gross et al.,
2016a; Palchan et al., 2018). The volcanic ash analog was collected from Mount Etna (Sicily, Italy) two month
after the eruption of 21 February 2022. The ash was taken from the upper cable car station "Funivia dell'Etna"
(37°704N, 14°999E). The samples were then processed through a setup of sieves to achieve a particle size smaller
than $63\mu m$ that are considered windblown (Guieu et al., 2010). The chemical and mineralogical properties of the
dust analogs are presented in Table 1.
To mimic dust deposition which typically occurs during a few major desert storms or volcanic eruption each year,
we applied the dust in two equivalent doses between 35-42 days after germination. Total application mass was 3
g per plant, to simulate the total dust deposition per $m^2$ for an average growth period in southern Israel (Gross et
al., 2021b). Dust treatments were done either directly on the foliage while covering the pot, preventing the dust
from touching the roots, or directly on the roots where the pots were subsequently covered with nylon to equalize
conditions with the foliage treated plants. Afterwards, the plants were left undisturbed with the settled dust
particles on their foliage or surface of the root area.

**Plant biomass and elemental analysis**
After harvesting, the plants were separated for roots and shoots, washed in 0.1M HCl and rinsed three times in
distilled water to remove dust particle residue (Gross et al., 2021; Lokshin et al., 2024a). To ensure that the
washing procedure removed all the applied dust particles from the leaf surfaces, we scanned surfaces of randomly
selected dusted and washed leaves with SEM-EDS which combines scanning electron microscope and energy-
dispersive X-ray spectroscopy to detect and analyze materials. After washing, plant tissue was dried, weighed and
root and shoot biomass were recorded. Afterwards, the dry shoot material was ground to powder and dry ashed at
550 C° in a furnace for four hours (Tiwari et al., 2022). Approximately 1g of the ashed material was subsequently
dissolved using 1 mL concentrated $HNO_3$ to achieve a clear solution. To prepare the dust types for elemental
analysis, the samples were dissolved on a hotplate by sequential dissolution using concentrated $HNO_3$, HF, and
HCl, resulting in clear solutions (Palchan et al., 2018). The elemental composition of the plants, dusts and nutrient
solution were analyzed at the Hebrew University using ICP-MS (Agilent 8900cx; Agilent Technology). Prior to
analysis, the ICP-MS was calibrated with a series of multi-element standard solutions (1 pg/mL - 100 ng/mL
Merck ME VI) and standards of major metals (300 ng/ml - 3 mg/mL). Internal standard (50 ng/mL Sc and 5 ng/mL
Re and Rh) was added to every standard and sample for drift correction. Standard reference solutions (USGS SRS

`



T-207, T-209) were examined at the beginning and end of the calibration to determine accuracy. The calculated
accuracies for the major and trace elements are 3% and 2%, respectively.
**Leaf surface pH**
Leaf surface pH was measured by manually attaching a portable pH electrode designed for flat surfaces (HI-1413;
HANNA pH instruments) onto the surface of three leaves from each plant. The measurements were performed
four times throughout the growing season (19, 24, 35 and 40 DAG) in the morning, two hours after sunrise.

**Trichome density**
Trichome density was determined in four young, fully developed leaves from four different plants per variety in
the P- treatment only (n=16). Leaves were scanned in a scanning electron microscope (VEGA3; Tescan, Czech
Republic). From each leaf, three photos of a 1mm$^2$ field were taken, and glandular and regular trichomes were
counted.

**Leaf exudates**
For analysis of the organic exudates, 2g of fresh leaves were sampled randomly from the P+ and P- treatments
before harvesting. The leaves were rinsed in 2 ml of distilled water and methanol (50:50) for 10 s. The extracted
surface metabolites were supplied with 50 µL of internal standard (ribitol, 0.2 mg ml$^{-1}$) and stored at -80°C until
analysis. Before analysis, the extracted samples were vacuum dried overnight at 35°C. The dried material was
redissolved in 40 µl of 20 mg mL$^{-1}$ methoxamine hydrochloride ($CH_3ONH_2$ HCl) in pyridine ($C_5H_5N$) and
derivatized for 90 min at 37°C, followed by a spike of 70 µL MSTFA (*N*-methyl-*N* (trimethylsilyl)
trifluoroacetamide ($CF_3CON(CH_3)Si(CH_3)_3$) at 37°C for 30 min**.** The dissolved metabolites were then introduced
to a mass spectrometry gas chromatograph (Agilent 6850 GC/5795C; Agilent Technology) for analysis. The
metabolites were detected by a mass spectrometer, where 1 µL of each sample was injected in split-less mode at
230°C to a helium carrier gas at a flow rate of 0.6 mL min$^{-1}$. GC processing was carried out using an HP-5MS
capillary column (30 m 9 0.250 mm 9 0.25 µm) and the spectrum was scanned for *m/z* 50–550 at 2.4 Hz. The ion
chromatograms and mass spectra obtained were evaluated using the MSD CHEMSTATION (E.02.00.493)
software, and sugars and amino acids were identified via comparison of retention times and mass spectra with
certified GC plant metabolite standards (Sigma Aldrich).

**Nd isotope chromatography and analysis**
Nd isotopes were measured on the dusts and in the above ground plant material at the end of the experiment. Nd
was extracted from the samples using TRU followed by LN-spec resins (Palchan et al., 2013). Measurements of
the isotopic ratios were performed using a Thermo Neptune multi-collector ICP-mass spectrometer at the
Weizmann Institute of Science. A JNdi Nd standard bracketed the samples, resulting with $^{143}$Nd/$^{144}$Nd value of
0.512035 ± 1$^{-5}$ (2$\sigma$, n=60). The data was normalized to $^{143}$Nd/$^{144}$Nd = 0.512115 (Tanaka et al., 2000). Rock
standards samples of BCR-2 were dissolved and measured along with the plant and dust samples yielding

`





$^{143}Nd/^{144}Nd$ value of $0.512628 \pm 6$ $(2\sigma)$ that agrees with $^{143}Nd/^{144}Nd = 0.512637 \pm 13$ value of BCR-2 (n=3)(Jweda
et al., 2016) . The Nd isotopic ratio is expressed as:

$$\varepsilon Nd = \left( \frac{\left( ^{143}Nd \big/ _{144}Nd \right)_{Sample}}{\left( ^{143}Nd \big/ _{144}Nd \right)_{CHUR}} - 1 \right) * 10,000$$

where the present value of $^{143}Nd/^{144}Nd = 0.512638$ in CHUR (Wasserburg et al., 1981). A sample isotopic
characterization is given in SI Table 4. The percent contribution of Nd within the leaves that comes either from
desert dust or volcanic ash (foliar contribution) was calculated using simple mixing equation of two components:

$$\% \, Foliar \, contribution = \frac{\varepsilon Nd_{sample} - \varepsilon Nd_{control}}{\varepsilon Nd_{end \, member} - \varepsilon Nd_{control}} * 100$$

Where $\varepsilon Nd_{sample}$ refers to plants that were treated either with desert dust or volcanic ash with, $\varepsilon Nd_{control}$ refers to
the untreated control plants and $\varepsilon Nd_{endmember}$ are the measured end member values of -10.3 for desert dust or 4.5
value for volcanic ash (Table SI-4 & Fig. 4).

**Mineralogical analysis**
Mineralogical composition of the dusts was determined with an X ray powder diffraction (XRD) using a
Panalytical Empyrean Powder Diffractometer equipped with a position sensitive X'Celerator detector. Cu K$\alpha$
radiation (k = 1.54178_A) at 40 kV and 30 mA. Scans were done over a 2h period, between 5° and 65° with an
approximate step size of 0.033°.

**Statistical Analysis**
Treatment comparisons for all measured parameters were tested using post-hoc Tukey honest significant
difference (HSD) tests (P < 0.05). The significant differences are denoted using different letters in the figures.
The standard errors of the mean in the vertical bars (in the figures) were calculated using GraphPad Prism version

241 9.0.0.


**Results**
**Plant biomass and total P under aCO$_2$ and eCO$_2$**
P starvation did not reduce P concentration in shoots but rather decreased shoot biomass gain. In addition, eCO$_2$
had no impact on P concentration or shoot biomass gain in the control -P plants, but significantly increased shoot

`



biomass gains in +P treated plants (Table 1). Thus, the treatment effects are reflected by changes in total plant P
(concentration multiplied by shoot biomass). The impact of desert and volcanic dust application on plants' foliage
was reflected by the increase of their total P content through shoot biomass gain rather than through changes in
shoot P concentration. Under $aCO_2$ conditions, desert dust application resulted in shoot biomass and total P content
increases of 35% and 21%, respectively, and volcanic ash application resulted in 28% and 35% increases,
respectively (Fig. 1 d,f). The root-treated plants did not show any increases in the shoot biomass or total P content
(Fig. 1 c,e). These trends are also seen in the $eCO_2$ conditions of 850 ppm atmospheric $CO_2$ experiment. Desert
dust application resulted in shoot biomass and total P content increases of 29% and 20%, respectively, and
volcanic ash application resulted in 62% and 51% shoot biomass increases, respectively (Fig. 2 d, f). Similarly,
the root-treated plants did not show any increases in the shoot biomass or total P content (Fig. 2 c, e). Unlike the
shoots, no significant changes of the biomass of the roots were detected across all treatments, thus changes in the
root shoot ratio reflect variations in shoot biomass rather that root biomass (Table 1).

**Root treatment 412 ppm**    **Foliar treatment 412 ppm**

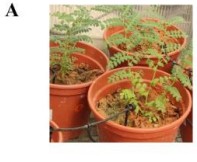
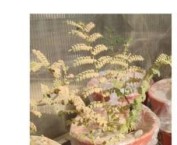

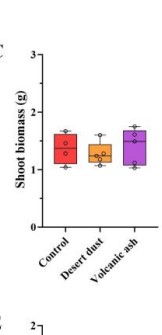
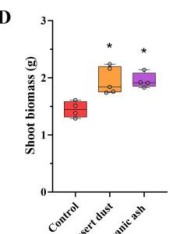

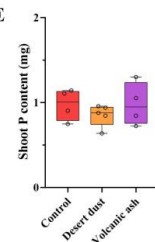
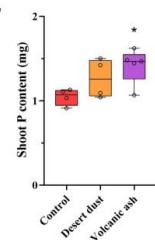


**Fig. 1** Biomass and total P content increases due to dust application treatments at $aCO_2$ of 412ppm. (a) Image of experiment
setting of the root treatment. (b) Image of experiment setting of foliar treatment. (c) Shoot biomass of root treated plants. (d)
Shoot biomass of foliar treated plants. (e) Shoot total P content of root treated plants. (f) Shoot total P content of foliar treated
plants. The asterisk denotes statistically significant difference from the control. The biomass and total P content in the root





treated plants do not show increases compared with the control groups. However, the foliar treatment of both desert dust and
volcanic ash caused significant increases in the shoot biomass and total P content. This implies that plants acquire P from fresh
dust deposits on their foliage and not from the root system. Red color represents control plants, orange desert dust treatment
and purple volcanic ash treatment.

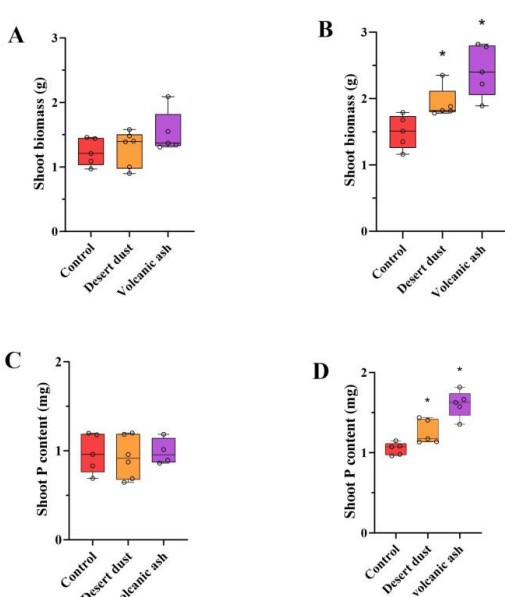


**Fig. 2** Biomass and total P content increases due to dust application treatments at $eCO_2$ of 850ppm. (a) Shoot biomass of root
treated plants. (b) shoot biomass of foliar treated plants. (c) Shoot total P content of root treated plants. (d) Shoot P content of
foliar treated plants. The asterisk denotes statistically significant difference from the control. The biomass and total P content
in the root treated plants do not show increases compared with the control groups. However, the foliar treatment of both desert
dust and volcanic ash caused significant increases in the shoot biomass and total P content. This implies that plants acquire P
from fresh dust deposits on their foliage and not from the root system. Red color represents control plants, orange desert dust
treatment and purple volcanic ash treatment.
**Elemental analysis of the plants**
The concentrations of selected micro and macro nutrients that build plants ionome, together with plants shoot
biomass, are given in Table 1.





**Table 1** Total elemental analysis of the plants (*Cicer arietinum cv 'Zehavit'*) fertilizers and dusts (ICP-MS analysis). The concentration of the different micro and macro elements are shown in ppm and plant biomass in g.

| Plant material (ppm) | Shoot biomass (g) | Root biomass (g) | Root/Shoot ratio | Mg | P | K | Ca | Mn | Fe | Ni | Cu | Zn |
|---|---|---|---|---|---|---|---|---|---|---|---|---|
| Control -P 412 #1 | 1.03 | 0.91 | 0.89 | 2749 | 715 | 21928 | 7257 | 52 | 75 | 1.3 | 2.9 | 22.9 |
| Control -P 412 #2 | 1.29 | 0.95 | 0.74 | 2828 | 860 | 21147 | 7266 | 45 | 112 | 2.3 | 10.8 | 23.5 |
| Discarded plant | | | | | | | | | | | | |
| Control -P 412 #4 | 1.51 | 1.36 | 0.90 | 2814 | 686 | 23832 | 7462 | 33 | 97 | 2.4 | 3.7 | 21.9 |
| Control -P 412 #5 | 1.38 | 1.22 | 0.88 | 2863 | 663 | 21883 | 7684 | 38 | 94 | 1.0 | 2.7 | 23.7 |
| Control -P 412 #6 | 1.61 | 1.39 | 0.87 | 2513 | 704 | 19705 | 6531 | 19 | 69 | 1.3 | 3.3 | 23.4 |
| Control -P 850 #1 | 1.35 | 1.02 | 0.75 | 2585 | 727 | 23099 | 6323 | 22 | 71 | 1.0 | 1.9 | 23.4 |
| Control -P 850 #2 | 1.16 | 1.02 | 0.88 | 3848 | 827 | 27768 | 7922 | 61 | 79 | 0.2 | 2.8 | 38.9 |
| Control -p 850 #3 | 1.79 | 1.39 | 0.78 | 2785 | 607 | 20121 | 7118 | 50 | 67 | 1.2 | 2.5 | 24.9 |
| Discarded because plant did not grow/withered | | | | | | | | | | | | |
| Control -P 850 #5 | 1.51 | 1.25 | 0.82 | 2847 | 759 | 27272 | 7572 | 21 | 88 | 2.2 | 4.7 | 26.4 |
| Control -P 850 #6 | 1.68 | 1.44 | 0.86 | 3180 | 640 | 24460 | 8732 | 31 | 93 | 1.6 | 2.9 | 29.1 |
| desert dust foliar-trated 412 ppm #1 | 2.24 | 1.65 | 0.74 | 2490 | 1458 | 23743 | 7040 | 47 | 125 | 0.9 | 2.6 | 21.9 |
| desert dust foliar-trated 412 ppm #2 | 1.74 | 1.44 | 0.83 | 2450 | 628 | 19416 | 6715 | 29 | 102 | 0.6 | 2.0 | 19.2 |
| desert dust foliar-trated 412 ppm #3 | 1.76 | 1.57 | 0.90 | 2326 | 855 | 17424 | 6576 | 27 | 97 | 1.0 | 2.3 | 20.5 |
| desert dust foliar-trated 412 ppm #4 | 2.16 | 1.87 | 0.87 | 2224 | 658 | 17576 | 6060 | 28 | 101 | 1.1 | 3.3 | 23.2 |
| desert dust foliar-trated 412 ppm #5 | 1.84 | 1.33 | 0.72 | 2611 | 566 | 21928 | 6817 | 40 | 116 | 1.1 | 3.0 | 20.3 |
| Discarded because plant did not grow/withered | | | | | | | | | | | | |
| desert dust foliar-treated 850 ppm #1 | 1.82 | 1.18 | 0.65 | 2274 | 626 | 21092 | 6599 | 26 | 151 | 2.2 | 2.8 | 22.1 |
| desert dust foliar-treated 850 ppm #2 | 1.78 | 1.58 | 0.89 | 2083 | 808 | 20320 | 5877 | 34 | 125 | 1.9 | 3.7 | 17.1 |
| Discarded because plant did not grow/withered | | | | | | | | | | | | |
| desert dust foliar-treated 850 ppm #4 | 2.35 | 1.52 | 0.65 | 2182 | 482 | 20380 | 7336 | 43 | 135 | 2.2 | 3.4 | 18.6 |
| desert dust foliar-treated 850 ppm #5 | 1.81 | 1.57 | 0.87 | 2995 | 648 | 24419 | 8366 | 39 | 169 | 2.4 | 3.5 | 25.0 |
| desert dust foliar-treated 850 ppm #6 | 1.88 | 1.92 | 1.02 | 2848 | 749 | 24303 | 8087 | 39 | 144 | 3.4 | 3.2 | 21.1 |
| volcanic ash foliar-treated 412 ppm #1 | 1.91 | 1.44 | 0.75 | 2499 | 755 | 20825 | 6058 | 34 | 137 | 0.6 | 3.0 | 18.1 |
| volcanic ash foliar-treated 412 ppm #2 | 2.14 | 1.74 | 0.81 | 2655 | 691 | 22032 | 6993 | 50 | 317 | 1.5 | 3.0 | 25.7 |
| volcanic ash foliar-treated 412 ppm #3 | 1.41 | 1.00 | 0.71 | 2524 | 757 | 18830 | 8067 | 49 | 148 | 1.0 | 2.9 | 20.8 |
| Discarded because plant did not grow/withered | | | | | | | | | | | | |
| volcanic ash foliar-treated 412 ppm #5 | 1.83 | 1.30 | 0.71 | 2814 | 800 | 23818 | 7121 | 40 | 177 | 1.4 | 3.5 | 23.7 |
| volcanic ash foliar-treated 412 ppm #6 | 1.92 | 1.49 | 0.78 | 2811 | 844 | 23122 | 7359 | 47 | 162 | 1.1 | 3.5 | 23.0 |
| Discarded because plant did not grow/withered | | | | | | | | | | | | |
| volcanic ash foliar-treated 850 ppm #2 | 2.22 | 1.85 | 0.83 | 2289 | 818 | 22549 | 6623 | 34 | 149 | 0.6 | 3.0 | 19.9 |
| volcanic ash foliar-treated 850 ppm #3 | 2.82 | 2.48 | 0.88 | 2365 | 558 | 23525 | 6848 | 41 | 373 | 2.4 | 3.4 | 18.2 |
| volcanic ash foliar-treated 850 ppm #4 | 2.40 | 2.19 | 0.91 | 2717 | 692 | 25778 | 7020 | 60 | 173 | 1.0 | 3.5 | 26.1 |
| volcanic ash foliar-treated 850 ppm #5 | 1.89 | 1.38 | 0.73 | 2584 | 718 | 24440 | 6722 | 43 | 140 | 0.7 | 3.0 | 23.5 |
| volcanic ash foliar-treated 850 ppm #6 | 2.78 | 2.37 | 0.85 | 2689 | 585 | 21384 | 7224 | 59 | 181 | 0.4 | 3.3 | 27.6 |
| Control +P 412 #1 | 10.46 | 3.33 | 0.32 | 5138 | 2465 | 30660 | 9429 | 79 | 161.3 | 1.1 | 5.1 | 51.2 |
| Control +P 412 #2 | 11.69 | 5.07 | 0.43 | 3729 | 2101 | 26096 | 7892 | 49 | 111.1 | 0.4 | 4.6 | 37.2 |
| Control +P 412 #3 | 11.47 | 4.44 | 0.39 | 6540 | 2148 | 29291 | 9076 | 69 | 88.9 | 0.7 | 4.3 | 43.5 |
| Control +P 412 #4 | 10.06 | 3.39 | 0.34 | 3322 | 1982 | 23933 | 6871 | 36 | 82.0 | 0.4 | 3.8 | 28.4 |
| Control+P 412 #5 | 10.76 | 3.94 | 0.37 | 3415 | 1804 | 23800 | 6970 | 44 | 95.5 | 0.4 | 4.1 | 33.1 |
| Control +P 412 #6 | 10.02 | 3.88 | 0.39 | 5147 | 2240 | 27966 | 8384 | 50 | 95.0 | 0.5 | 4.6 | 38.5 |
| Discarded plant | | | | | | | | | | | | |
| Control +P 850 #2 | 13.40 | 7.24 | 0.54 | 3759 | 2253 | 26837 | 7886 | 59 | 91.4 | 0.8 | 3.8 | 32.6 |
| Discarded plant | | | | | | | | | | | | |
| Control +P 850 #4 | 17.17 | 7.29 | 0.42 | 3202 | 2196 | 25021 | 8052 | 68 | 96.6 | 0.8 | 5.9 | 30.5 |
| Control+P 850 #5 | 17.51 | 10.85 | 0.62 | 3633 | 2258 | 27403 | 8860 | 67 | 97.0 | 0.7 | 4.1 | 31.7 |
| Control +P 850 #6 | 15.86 | 6.55 | 0.41 | 5488 | 2959 | 30362 | 11394 | 100 | 109.5 | 0.9 | 4.9 | 50.5 |
| **Fertilizers and dusts (ppm)** | | | | | | | | | | | | |
| +P fertilizer | | | | 1226 | 713 | 6000 | 11 | 76 | 151 | 0.4 | 5.6 | 50.1 |
| -P fertilizer | | | | 1214 | 35 | 7808 | 7 | 70 | 136 | 0.4 | 5.1 | 47.5 |
| Desert dust | | | | 6513 | 1387 | 8673 | 136081 | 245 | 12745 | 18.0 | 10.0 | 39.0 |
| Volcanic ash #1 | | | | 23534 | 1669 | 12514 | 64461 | 1097 | 63736 | 49.5 | 118.6 | 80.0 |
| Volcanic ash #2 | | | | 22648 | 1788 | 12056 | 61628 | 1066 | 61903 | 48.3 | 115.6 | 73.7 |

**Physiological adaptations toward foliar uptake**

The domesticated variety 'Zehavit' showed a strong response to the foliar treatment with up 35% increased biomass compared to the control group, whereas the wild variety CR934 showed up to 5% increases compared with the control group (Fig. 3a). The leaf pH of the Zehavit was 1.15 and of the CR934 it was 2.7 (Fig. 3b),



trichome density, both glandular and non-glandular, were higher in the Zehavit compared to the CR934 (Fig. 3c-
e). The exudates of oxalic, malic, and citric acids were significantly higher at the Zehavit in comparison to CR934
(Fig. 3f). The results indicate increased biomass, lower pH, higher trichome density, and higher exudate levels in
the 'Zehavit' variety.





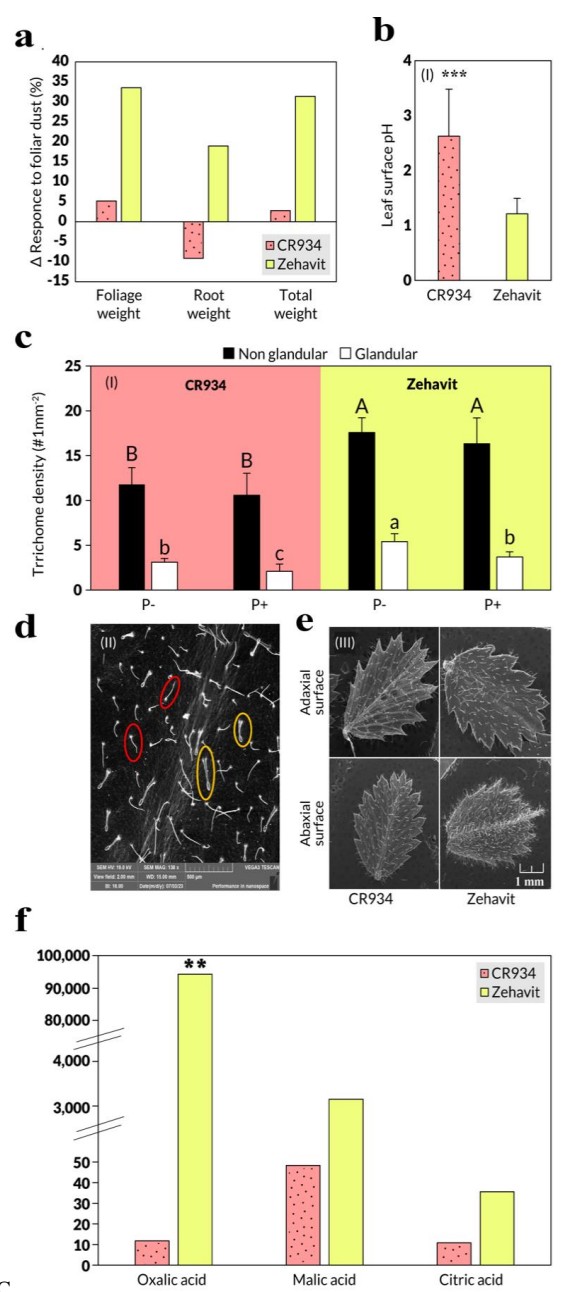

**Fig. 3** Comparison of two chickpea varieties - CR934 (dotted, pink) and Zehavit (yellow) and their leaf properties under dust foliar fertilization. (a) Biomass and P uptake response to foliar dust P. Each column indicates the difference Δ (%) between the foliar dusted plants and the control untreated plants (n=6). (b) (I) Leaf surface pH. Each value indicates an average of five measurements on a plant throughout the growth season in control treatment (n=90), and two measurements in foliar dust treatment (n=10). One asterisk indicates significant differences between treatments using a T-test, and a one-way ANOVA (P≤0.05). Three asterisks indicate significant differences between treatments using a T-test, and a one-way ANOVA (P≤0.001). (c) Leaf non-Glandular (black column) and glandular (white column) trichrome density in CR934 and Zehavit control plants (-P and +P). Different letters indicate significant differences between varieties and treatments using Tukey-HSD test (P≤0.05) (n=12). Capital letters refer to non-glandular trichomes and small letters refer to glandular trichomes. (d). SEM scans of non-glandular (red circles) and glandular (yellow circles) trichomes of typical Zehavit leaf. (e). SEM scans of leaves of CR934 (left) and Zehavit (right) varieties. The Zehavit clearly shows higher density of trichomes in the abaxial surface, rendering it as more fit to extract nutrients from dust particles. (f). Exudates of organic acids. Each column indicates the average of leaf washing from four plants, in P- control treatment (n=4). Two asterisks indicate significant differences between treatments using a T-test, and a one-way ANOVA (P≤0.01). Values are concentrations compared with an internal standard.

C

**Nd isotopic analysis of the dusts, control plants and the treated plants**

We utilized the ratio of $^{143}Nd/^{144}Nd$ in the εNd notation to trace the source of Nd in our experiments and quantify
the flux of dust-borne Nd to plants as an indirect measure of dust nutrient transfer (Fig. 4). The mineral dust



analogues presented εNd values of 5 and -11 for the volcanic ash and the desert dust, respectively. Plant material
εNd values of the control plants, that reflect the inheritance value (i.e., arising from the seed Nd isotope
composition) was -0.3, desert dust treated plants were characterized with values of -8.8 to -5, and the volcanic ash
treated plants were characterized with values of 3.4 to 4. Both treated plant groups are significantly different than
the inheritance value of -0.3 characterizing the control group.






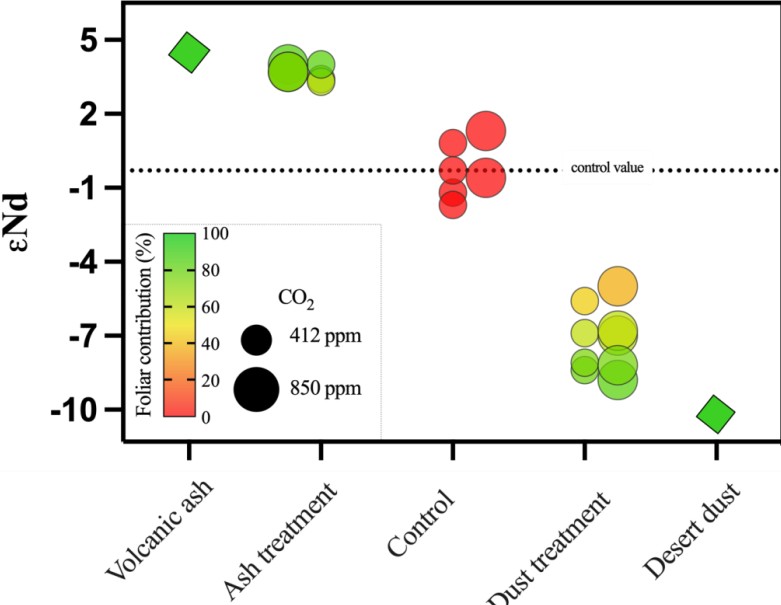


**Fig. 4** Quantification of dust mineral-nutrient flux from the foliage. Radiogenic isotopic ratios of $^{143}Nd/^{144}Nd$ in the different sample groups (x-axis) expressed in $\varepsilon Nd$ values. Diamonds represent the two applied mineral fractions of volcanic ash and desert dust; circles represent plants treated with the desert and volcanic dusts and the control groups. Large circles represent plants growing in the 850 ppm $eCO_2$ and small circles represent the 412 ppm $aCO_2$. The color scale reflects the % contribution of Nd originating from the dusts via the foliage, which was calculated using a two-component mixing model. The control plants' Nd signature reflects the inheritance value from the seed, where a value of $\varepsilon Nd$=-0.3 is set as the control, $\varepsilon Nd$=-10.3 as the desert dust value, and $\varepsilon Nd$=4.6 as the volcanic ash value. A foliar contribution of more than 60% is evident in the plants applied with desert dust and more than 70% in the plants applied with volcanic ash. Standard errors on the isotopic values are all smaller than the depicted data points.

**Discussion**

**Foliar mineral-nutrients uptake**

In our experiments, we simulated desert dust and volcanic ash deposition by manually applying them on chickpea plants (*Cicer arietinum cv 'Zehavit'*). The dust was applied separately either on the surface of the pot near the roots, or on its foliage (Fig. 1), while control plants were not treated with dust. After several weeks, a significant impact of the foliar treatment was already noticeable where shoot biomass and total P content in the foliage-treated plants had increased, following dusts treatment, compared with the control group. In contrast, the root-treated plants did not show any increases in the biomass or P content, suggesting that over short timescales (i.e., several



weeks), foliar uptake is the only nutrient uptake pathway from freshly deposited dust (Fig. 1c, e). These results
were then replicated when a similar experiment was conducted with plants grown on sandy soil, in bigger pots
(Fig. S1), emphasizing that our observations are not limited to the specific artificial experimental conditions in
perlite (which may bias root behavior), but also apply for real soil conditions (Fig. S1).
**Plant strategies for foliar mineral-nutrient uptake**
Most of the P in the dusts is incorporated in the mineral lattice of minerals such as apatite (Dam et al., 2021),
which is largely insoluble under the natural rhizosphere pH range (Hinsinger, 2001)(Hinsinger, 2001). Hence, P
in volcanic or desert dust has low bioavailability for root uptake as was also shown in Lokshin et al, (2024a) with
fire ash. On the leaf surface however, chemical, morphological, and microbial modifications may promote nutrient
solubility and bioavailability and thus enable uptake through the leaf surface (Gross et al., 2021; Muhammad et
al., 2019)(Gross et al., 2021; Muhammad et al., 2019). Examining two chickpea varieties with contrasting
responses to dust application: wild variety CR934, and common domesticated variety Zehavit, we found a few
properties that facilitate foliar P acquisition from dust (Fig. 3). These include structural, morphological, and
chemical modifications that are comparable to those reported in the rhizosphere (Hinsinger, 2001)(Hinsinger,
2001). The foliar-uptake-efficient variety Zehavit has significantly more acidic leaf surface (pH ~ 1, Fig. 3b), and
thus promotes both dissolution and mobility of P from the pH sensitive mineral apatite (Gross et al., 2015)(Gross
et al., 2015), as well as other mineral-nutrients in the dust (Bradl, 2004; Gross et al., 2021; Muhammad et al.,
2019)(Bradl, 2004; Gross et al., 2021; Muhammad et al., 2019). Additionally, a unique set of metabolites secreted
from the leaf surface further facilitated the foliar uptake pathway in Zehavit. These include increased
concentrations of oxalate and malate, which are known to release insoluble P in soils through anion exchange
reactions (Lambers et al., 2019; Tiwari et al., 2022)(Lambers et al., 2019; Tiwari et al., 2022), and increased levels
of sugars such as glucose and sucrose that may promote the activity of nutrient solubilizing microbes on the
phyllosphere (Shakir et al., 2021)(Shakir et al., 2021) (Fig. 3f, fig. S2). We further found that Zehavit showed
higher leaf trichome density on both leaf axial and adaxial sides (Fig. 3 c,d,e). These trichomes facilitate the
release of metabolites and promote adhesion of dust captured on leaf surfaces (fig. S3) (Gross et al., 2021)(Gross
et al., 2021). We postulate that other plant species share comparable leaf traits that enhance dust capture and
solubility such as wheat and various tree species that showed strong responses to foliar dust fertilization (Gross et
al., 2021; Starr et al., 2023)(Gross et al., 2021; Starr et al., 2023). Overall, our results suggest that the combination
of leaf surface acidification, secretion of organic acids and additional exudations combined with an increased
trichome density enhances foliar dust capture and nutrient uptake in chickpeas. Results of previous study with
application of inert silicon powder on chickpea leaf surface indicate that the shading effect resulting leaf surface
coverage with dust has low effect on plant growth and photosynthesis (Gross et al. (2021). Yet, the dust shading
effect was more pronounced in several tree species (Starr *et al.*, 2023), suggesting the contrasting impact of
coverage of the foliage should be considered.
**Dust impact on plant nutrient status under eCO$_2$**
Numerous studies reported that eCO$_2$ conditions reduce the concentrations of several nutrients in plant tissues
such as Fe, Zn, Cu, Mn, Ni and others (Loladze, 2002; Fernando et al., 2014; Myers *et al.*, 2014; Gojon *et al.,*
2023). The reduction in shoot nutrient concentrations was also observed in our experiments (fig. 5). In accordance



with previous knowledge (Loladze, 2002)(Loladze, 2002), plants that were grown under eCO$_2$ in our experiment
showed a significant reduction of 10-50% in the concentrations of nutrients such as Mg, K, Ca, Mn, Zn and Fe,
with even more significant reductions in Cu and Ni (72% and 90%, respectively), (Fig. 5). Although we did not
observe statistically significant differences in biomass between control plants grown under aCO$_2$ and eCO$_2$
conditions (P = 0.4), the reduction in essential macro- and micronutrient concentrations may be partly explained
by the effect of nutrient dilution. Another potential reason for the nutrient decline under eCO$_2$ could be related to
reduced efficiency in mineral nutrient absorption through the root system (Gojon et al., 2023).Click or tap here
to enter text.Click or tap here to enter text.. We found that foliar application of both volcanic and desert
dust on plants that were grown under eCO$_2$ replenished their Fe and Ni concentrations (both essential
micronutrients for plant growth and in the human diet) compared with the control group (fig, 5a,b). Desert dust
treated plants showed increases of Fe and Ni concentrations of 44% and 46%, respectively (Fig. 5a). Volcanic ash
treated plants showed Fe elevated concentrations of 66% (Fig. 5b). The Ni concentrations had more moderate
increases from volcanic ash, with 40% higher than in the aCO$_2$. These increases returned Fe and Ni back to
standard, nontoxic levels (Shahzad et al. 2018). These results emphasize that the role foliar uptake of atmospheric
nutrients on the mineral nutrition level of plants will be greater under eCO$_2$ and offset the projected nutrient
reduction driven by the dilution effect and the downregulation of the root's nutrient uptake pathway (Zhu et al.,
2018)(Zhu et al., 2018).


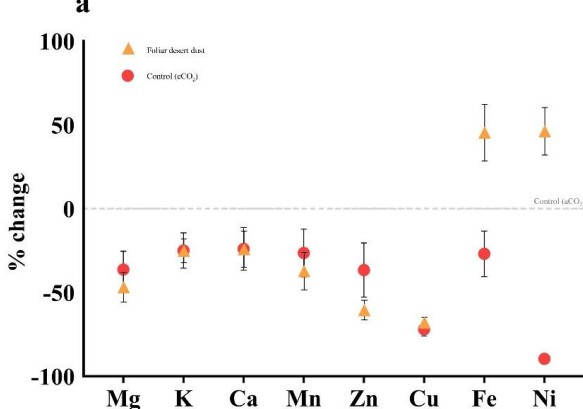





397

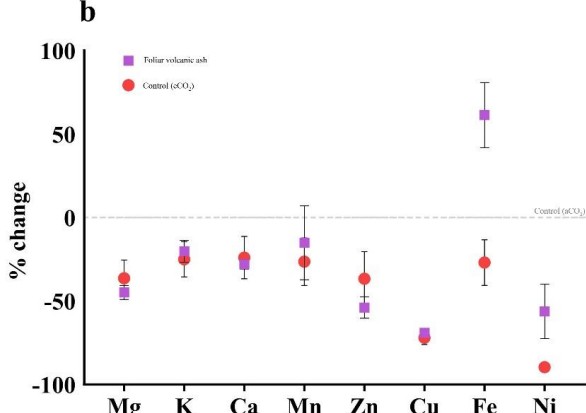

**Fig. 5** Comparison of the % change in plant nutrient concentration under $eCO_2$ compared with $aCO_2$ control plants. The comparison was conducted as follows: the average value of each nutrient in plants grown under aCO2 was calculated, and then each nutrient in individual chickpea plants grown under eCO2 levels was expressed as a ratio relative to the average under aCO2 conditions ($eCO_2$ plant (each individual plant) /$aCO_2$ plant (average of all the control plants)). Changes in nutrient concentrations of the control $eCO_2$ plants (red circles) show that $eCO_2$ conditions deteriorate plant nutritional status significantly. (a) The effect of foliar treatment of desert dust (orange triangles). (b) The effect of foliar treatment of volcanic ash (purple squares). Error bars denote SD.

**Quantifying the contribution of foliar nutrient uptake from dust**

Traditionally, radiogenic Nd isotopes serve as excellent tracers for sources of magmatic rocks (Stein and Goldstein, 1996)(Stein and Goldstein, 1996), sediment archives (Chadwick et al., 1999; Palchan et al., 2018)(Chadwick et al., 1999; Palchan et al., 2018), and water bodies (Farmer et al., 2019)(Farmer et al., 2019). Since Nd is found in high concentration in nutrient bearing minerals (Aciego et al., 2017; Arvin et al., 2017; Chadwick et al., 1999)(Aciego et al., 2017; Arvin et al., 2017; Chadwick et al., 1999),  Nd isotopes were recently used to trace P sources in plant tissues, where it was shown that the contribution of dust outpaces the weathering of the local bedrock over geological time scales (Aciego et al., 2017; Arvin et al., 2017)(Aciego et al., 2017; Arvin et al., 2017). While the use of Nd isotopes to other elements such as P provides new knowledge on their sources, it should be done cautiously because different elements have differing speciation, uptake mechanisms, and transport kinetics in plant tissue. Here, we utilized the ratio of $^{143}Nd/^{144}Nd$ in the εNd notation to trace the source of Nd in our experiments and quantify its flux to plant tissue from dust. From this measurement we can approximate the flux of P, Fe and Ni via foliar pathway (Fig. 4). We used a two-component mixing model, where the average εNd value of the control plants, -0.3, which arise from the Nd "inheritance" (i.e., the Nd composition of the seed) is regarded as one end member, and dust εNd values are regarded as the second end member, with values of -11 (desert dust) and 5 (volcanic ash). We found that desert dust treated plants were characterized with εNd values of -8.8 to -5, significantly different than the inheritance value of the control group. Similarly, the



volcanic ash treated plants were characterized with εNd values of 3.4 to 4, significantly different than the
inheritance value of -0.3. Thus, it is evident that the εNd of the foliage-treated plants comprise a mixture of the
inheritance and the type of dust applied. Based on the mixing model, the chickpea plant acquired over 60% of its
Nd from desert dust deposited on the foliage. Volcanic ash deposited on the foliage contributed over 70% of its
Nd (Fig. 4). However, Nd isotopes do not show the increased supplement of Fe and Ni in plants that were grown
under $eCO_2$. Thus, more data on the relation between Nd and other nutrients uptake will advance its use in future
studies to quantify the immediate contribution of freshly deposited dust on plants nutrition in field and lab
experimental settings.

In conclusion, we showed here that dust nutrient uptake via the foliar pathway in chickpea plants plays a major
role in their nutrition. Plant foliage captures and dissolves freshly deposited dust particles, making atmospheric
mineral nutrients more accessible through the foliage on a short time scale than via the roots. Most of the P in the
dust is incorporated in the mineral lattice of minerals such as apatite (Dam et al., 2021)(Dam et al., 2021), which
is largely insoluble under the natural rhizosphere pH range (Hinsinger, 2001)(Hinsinger, 2001). Hence, P in dust
has low bioavailability for root uptake. On the leaf surface however, chemical, morphological, and microbial
modifications may promote nutrient solubility and bioavailability and facilitate uptake through the leaf surface
(Gross et al., 2021; Muhammad et al., 2019)(Gross et al., 2021; Muhammad et al., 2019). Thus, our findings
highlight that dust serves as an alternative source of nutrients to plants from the foliage on short timescales of a
few weeks. Furthermore, that foliar dust acquisition compensates for the reduction in nutrients such as Fe and Ni,
induced by $eCO_2$ conditions (Gojon et al., 2023)(Gojon et al., 2023). The broader aspect of our findings
emphasizes the central role of dust in plant nutrition through the foliar pathway and to global biogeochemical
cycles. Our findings imply that the foliar nutrient uptake pathway from natural dust will play a central role in
$eCO_2$ earth, and that this pathway may be a target for novel fertilization techniques to compensate for the expected
decline in the crops' nutritional value.



**Acknowledgments**

We thank Dr. Yigal Erel and Ofir Tirosh from the Hebrew University of Jerusalem for their support in ICP-MS
analyses, and Dr. Yael Kiro from Weismann Institute for conducting isotopic chromatography in her lab, and
Dr. Sthephen Fox for his support in MC-ICP-MS analyses.
**Author Contributions:**
Conceptualization: DP, AG, RE
Dust sampling: AL, DA, AG
Methodology: DP, AG, RE
Investigation: AL, EG, SF



468 Visualization: DP, AL, EG

469 Funding acquisition: AG, RE, AL

470 Project administration: DP, AG

471 Supervision: DP, AG, RE

472 Writing – original draft: DP, AG, AL, RE

473 **Competing Interest Statement:** The authors declare no competing interests.

474 **Classification:** Physical Sciences - Earth, Atmospheric, and Planetary Sciences; Biological Sciences - Plant
475 Biology.

476

477

478 **References**

479 Aciego, S. M., Riebe, C. S., Hart, S. C., Blakowski, M. A., Carey, C. J., Aarons, S. M., Dove, N. C.,
480 Botthoff, J. K., Sims, K. W. W., and Aronson, E. L.: Dust outpaces bedrock in nutrient supply to
481 montane forest ecosystems, Nat Commun, 8, 14800, https://doi.org/10.1038/ncomms14800,
482 2017.

483 Arvin, L. J., Riebe, C. S., Aciego, S. M., and Blakowski, M. A.: Global patterns of dust and
484 bedrock nutrient supply to montane ecosystems, Sci Adv, 3, eaao1588,
485 https://doi.org/10.1126/sciadv.aao1588, 2017.

486 Bauters, M., Drake, T. W., Wagner, S., Baumgartner, S., Makelele, I. A., Bodé, S., Verheyen, K.,
487 Verbeeck, H., Ewango, C., Cizungu, L., Van Oost, K., and Boeckx, P.: Fire-derived phosphorus
488 fertilization of African tropical forests, Nat Commun, 12, https://doi.org/10.1038/S41467-021-
489 25428-3, 2021.

490 Bradl, H. B.: Adsorption of heavy metal ions on soils and soils constituents, J Colloid Interface
491 Sci, 277, 1–18, https://doi.org/10.1016/J.JCIS.2004.04.005, 2004.

492 Chadwick, O. A., Derry, L. A., Vitousek, P. M., Huebert, B. J., and Hedin, L. O.: Changing sources
493 of nutrients during four million years of ecosystem development, Nature, 397, 491–497,
494 https://doi.org/10.1038/17276, 1999.

495 Ciriminna, R., Scurria, A., Tizza, G., and Pagliaro, M.: Volcanic ash as multi-nutrient mineral
496 fertilizer: Science and early applications, JSFA Reports, 2, 528–534,
497 https://doi.org/10.1002/JSF2.87, 2022.

498 Clarkson, D. T. and Hanson, J. B.: THE MINERAL NUTRITION OF HIGHER PLANTS, Ann. Rev.
499 Plant Physiol, 31, 239–98, 1980.

500 Dam, T. T. N., Angert, A., Krom, M. D., Bigio, L., Hu, Y., Beyer, K. A., Mayol-Bracero, O. L.,
501 Santos-Figueroa, G., Pio, C., and Zhu, M.: X-ray Spectroscopic Quantification of Phosphorus
502 Transformation in Saharan Dust during Trans-Atlantic Dust Transport, Cite This: Environ. Sci.
503 Technol, 55, 12694–12703, https://doi.org/10.1021/acs.est.1c01573, 2021.

`



Eger, A., Almond, P. C., and Condron, L. M.: Phosphorus fertilization by active dust deposition
in a super-humid, temperate environment—Soil phosphorus fractionation and accession
processes, Global Biogeochem Cycles, 27, 108–118, https://doi.org/10.1002/GBC.20019,
507  2013.

Farmer, J. R., Hönisch, B., Haynes, L. L., Kroon, D., Jung, S., Ford, H. L., Raymo, M. E., Jaume-
Seguí, M., Bell, D. B., Goldstein, S. L., Pena, L. D., Yehudai, M., and Kim, J.: Deep Atlantic Ocean
carbon storage and the rise of 100,000-year glacial cycles, Nature Geoscience 2019 12:5, 12,
355–360, https://doi.org/10.1038/s41561-019-0334-6, 2019.
Gojon, A., Cassan, O., Bach, L., Lejay, L., and Martin, A.: The decline of plant mineral nutrition
under rising CO2: physiological and molecular aspects of a bad deal, Trends Plant Sci, 28, 185–
198, https://doi.org/10.1016/J.TPLANTS.2022.09.002, 2023.
Goll, D. S., Bauters, M., Zhang, H., Ciais, P., Balkanski, Y., Wang, R., and Verbeeck, H.:
Atmospheric phosphorus deposition amplifies carbon sinks in simulations of a tropical forest in
Central Africa, New Phytologist, 237, 2054–2068, https://doi.org/10.1111/NPH.18535, 2023.
Gross, A., Goren, T., Pio, C., Cardoso, J., Tirosh, O., Todd, M. C., Rosenfeld, D., Weiner, T.,
Custoio, D., and Angert, A.: Variability in Sources and Concentrations of Saharan Dust
Phosphorus over the Atlantic Ocean, Environ Sci Technol Lett, 2, 31–37,
https://doi.org/10.1021/ez500399z, 2015.
Gross, A., Palchan, D., Krom, M. D., and Angert, A.: Elemental and isotopic composition of
surface soils from key Saharan dust sources, Chem Geol, 442, 54–61,
https://doi.org/10.1016/j.chemgeo.2016.09.001, 2016a.
Gross, A., Turner, B. L., Goren, T., Berry, A., and Angert, A.: Tracing the Sources of Atmospheric
Phosphorus Deposition to a Tropical Rain Forest in Panama Using Stable Oxygen Isotopes,
Environ Sci Technol, 50, 1147–1156, https://doi.org/10.1021/ACS.EST.5B04936, 2016b.
Gross, A., Tiwari, S., Shtein, I., and Erel, R.: Direct foliar uptake of phosphorus from desert dust,
New Phytologist, 230, 2213–2225, https://doi.org/10.1111/nph.17344, 2021a.
Gross, A., Tiwari, S., Shtein, I., and Erel, R.: Direct foliar uptake of phosphorus from desert dust,
New Phytologist, 230, 2213–2225, https://doi.org/10.1111/NPH.17344, 2021b.
Guieu, C., Dulac, F., Desboeufs, K., Wagener, T., Pulido-Villena, E., Grisoni, J. M., Louis, F.,
Ridame, C., Blain, S., Brunet, C., Bon Nguyen, E., Tran, S., Labiadh, M., and Dominici, J. M.:
Large clean mesocosms and simulated dust deposition: A new methodology to investigate
responses of marine oligotrophic ecosystems to atmospheric inputs, Biogeosciences, 7, 2765–
2784, https://doi.org/10.5194/BG-7-2765-2010, 2010.
Hinsinger, P.: Bioavailability of soil inorganic P in the rhizosphere as affected by root-induced
chemical changes: A review, Plant Soil, 237, 173–195,
https://doi.org/10.1023/A:1013351617532/METRICS, 2001.
Jweda, J., Bolge, L., Class, C., and Goldstein, S. L.: High Precision Sr-Nd-Hf-Pb Isotopic
Compositions of USGS Reference Material BCR-2, Geostand Geoanal Res, 40, 101–115,
https://doi.org/10.1111/j.1751-908X.2015.00342.x, 2016.
Kok, J. F., Adebiyi, A. A., Albani, S., Balkanski, Y., Checa-Garcia, R., Chin, M., Colarco, P. R.,
Hamilton, D. S., Huang, Y., Ito, A., Klose, M., Li, L., Mahowald, N. M., Miller, R. L., Obiso, V.,
Pérez García-Pando, C., Rocha-Lima, A., and Wan, J. S.: Contribution of the world's main dust



source regions to the global cycle of desert dust, Atmos Chem Phys, 21, 8169–8193,
https://doi.org/10.5194/ACP-21-8169-2021, 2021.
Lal, R.: Soil degradation as a reason for inadequate human nutrition, Food Secur, 1, 45–57,
https://doi.org/10.1007/S12571-009-0009-Z, 2009.
Lambers, H., Albornoz, F. E., Arruda, A. J., Barker, T., Finnegan, P. M., Gille, C., Gooding, H.,
Png, K., Ranathunge, K., and Zhong, H.: Nutrient-acquisition strategies, in: A Jewel in the crown
of a global biodiversity hotspot. , edited by: Lambers, H., Kwongan Foundation and the Western
Australian Naturalists' Club Inc., Perth, 2019.
Van Langenhove, L., Verryckt, L. T., Bréchet, L., Courtois, E. A., Stahl, C., Hofhansl, F., Bauters,
M., Sardans, J., Boeckx, P., Fransen, E., Peñuelas, J., and Janssens, I. A.: Atmospheric
deposition of elements and its relevance for nutrient budgets of tropical forests, 149, 175–193,
https://doi.org/10.1007/s10533-020-00673-8, 2020.
Langmann, B.: Volcanic Ash versus Mineral Dust: Atmospheric Processing and Environmental
and Climate Impacts, ISRN Atmospheric Sciences, 2013, 1–17,
https://doi.org/10.1155/2013/245076, 2013a.
Langmann, B.: Volcanic Ash versus Mineral Dust: Atmospheric Processing and Environmental
and Climate Impacts, ISRN Atmospheric Sciences, 2013, 1–17,
https://doi.org/10.1155/2013/245076, 2013b.
Lokshin, A., Palchan, D., and Gross, A.: Direct foliar phosphorus uptake from wildfire ash,
Biogeosciences, 21, 2355–2365, https://doi.org/10.5194/BG-21-2355-2024, 2024a.
Lokshin, A., Gross, A., Dor, Y. Ben, and Palchan, D.: Rare earth elements as a tool to study the
foliar nutrient uptake phenomenon under ambient and elevated atmospheric $CO_2$
concentration, Science of The Total Environment, 948, 174695,
https://doi.org/10.1016/J.SCITOTENV.2024.174695, 2024b.
Loladze, I.: Rising atmospheric $CO_2$ and human nutrition: toward globally imbalanced plant
stoichiometry?, Trends Ecol Evol, 17, 457–461, https://doi.org/10.1016/S0169-5347(02)02587-
572    9, 2002.

Loladze, I.: Hidden shift of the ionome of plants exposed to elevated $CO_2$ depletes minerals at
the base of human nutrition, Elife, 2014, https://doi.org/10.7554/ELIFE.02245, 2014.
Longo, A. F., Ingall, E. D., Diaz, J. M., Oakes, M., King, L. E., Nenes, A., Mihalopoulos, N., Violaki,
K., Avila, A., and Benitez-Nelson, C. R.: P-NEXFS analysis of aerosol phosphorus delivered to
the Mediterranean Sea, Geophys Res Lett, 41, 4043–4049, 2014.
Lowe, N. M.: The global challenge of hidden hunger: perspectives from the field, Proceedings of
the Nutrition Society, 80, 283–289, https://doi.org/10.1017/S0029665121000902, 2021.
Marschner, H., Kirkby, E. A., and Engels, C.: Importance of Cycling and Recycling of Mineral
Nutrients within Plants for Growth and Development, Botanica Acta, 110, 265–273,
https://doi.org/10.1111/J.1438-8677.1997.TB00639.X, 1997.
Muhammad, S., Wuyts, K., and Samson, R.: Atmospheric net particle accumulation on 96 plant
species with contrasting morphological and anatomical leaf characteristics in a common
garden experiment, Atmos Environ, 202, 328–344,
https://doi.org/10.1016/J.ATMOSENV.2019.01.015, 2019.



Myers, S. S., Zanobetti, A., Kloog, I., Huybers, P., Leakey, A. D. B., Bloom, A. J., Carlisle, E.,
Dietterich, L. H., Fitzgerald, G., Hasegawa, T., Holbrook, N. M., Nelson, R. L., Ottman, M. J.,
Raboy, V., Sakai, H., Sartor, K. A., Schwartz, J., Seneweera, S., Tausz, M., and Usui, Y.:
Increasing CO2 threatens human nutrition, Nature 2014 510:7503, 510, 139–142,
https://doi.org/10.1038/nature13179, 2014.
Nakamaru, Y., Nanzyo, M., and Yamasaki, S. I.: Utilization of apatite in fresh volcanic ash by
pigeonpea and chickpea, Soil Sci Plant Nutr, 46, 591–600,
https://doi.org/10.1080/00380768.2000.10409124, 2000.
Okin, G. S., Mahowald, N., Chadwick, O. A., and Artaxo, P.: Impact of desert dust on the
biogeochemistry of phosphorus in terrestrial ecosystems, Global Biogeochem Cycles, 18,
https://doi.org/10.1029/2003GB002145, 2004.
Van Oss, R., Abbo, S., Eshed, R., Sherman, A., Coyne, C. J., Vandemark, G. J., Zhang, H. Bin, and
Peleg, Z.: Genetic relationship in cicer Sp. expose evidence for geneflow between the cultigen
and its wild progenitor, PLoS One, 10, https://doi.org/10.1371/journal.pone.0139789, 2015.
Palchan, D., Stein, M., Almogi-Labin, A., Erel, Y., and Goldstein, S. L.: Dust transport and
synoptic conditions over the Sahara–Arabia deserts during the MIS6/5 and 2/1 transitions from
grain-size, chemical and isotopic properties of Red Sea cores, Earth Planet Sci Lett, 382, 125–
139, https://doi.org/10.1016/j.epsl.2013.09.013, 2013.
Palchan, D., Erel, Y., and Stein, M.: Geochemical characterization of contemporary fine detritus
in the Dead Sea watershed, Chem Geol, 494, 30–42,
https://doi.org/10.1016/J.CHEMGEO.2018.07.013, 2018.
Shakir, S., Zaidi, S. S. e. A., de Vries, F. T., and Mansoor, S.: Plant Genetic Networks Shaping
Phyllosphere Microbial Community, https://doi.org/10.1016/j.tig.2020.09.010, 1 April 2021.
Starr, M., Klein, T., and Gross, A.: Direct foliar acquisition of desert dust phosphorus fertilizes
forest trees despite reducing photosynthesis, Tree Physiol, 43, 794–804,
https://doi.org/10.1093/TREEPHYS/TPAD012, 2023.
St.Clair, S. B. and Lynch, J. P.: The opening of Pandora's Box: climate change impacts on soil
fertility and crop nutrition in developing countries, 335, 101–115, https://doi.org/10.1007/s,
615 2010.

Stein, M. and Goldstein, S. L.: From plume head to continental lithosphere in the Arabian-
Nubian shield, Nature, 382, 773–778, 1996.
Stockdale, A., Krom, M. D., Mortimer, R. J. G., Benning, L. G., Carslaw, K. S., Herbert, R. J., Shi,
Z., Myriokefalitakis, S., Kanakidou, M., and Nenes, A.: Understanding the nature of atmospheric
acid processing of mineral dusts in supplying bioavailable phosphorus to the oceans,
Proceedings of the National Academy of Sciences, 113, 14639–14644, 2016.
Tanaka, T., Togashi, S., Kamioka, H., Amakawa, H., Kagami, H., Hamamoto, T., Yuhara, M.,
Orihashi, Y., Yoneda, S., Shimizu, H., Kunimaru, T., Takahashi, K., Yanagi, T., Nakano, T.,
Fujimaki, H., Shinjo, R., Asahara, Y., Tanimizu, M., and Dragusanu, C.: JNdi-1: a neodymium
isotopic reference in consistency with LaJolla neodymium, Chem Geol, 168, 279–281,
https://doi.org/10.1016/S0009-2541(00)00198-4, 2000.
Tiwari, S., Erel, R., and Gross, A.: Chemical processes in receiving soils accelerate
solubilisation of phosphorus from desert dust and fire ash, Eur J Soil Sci, 73,
https://doi.org/10.1111/EJSS.13270, 2022.



Wasserburg, G. J., Jacobsen, S. B., DePaolo, D. J., McCulloch, M. T., and Wen, T.: Precise
determination of Sm/Nd ratios, Sm and Nd isotopic abundances in standard solutions,
Geochim Cosmochim Acta, 45, 2311–2323, https://doi.org/10.1016/0016-7037(81)90085-5,
633  1981.

Zhu, C., Kobayashi, K., Loladze, I., Zhu, J., Jiang, Q., Xu, X., Liu, G., Seneweera, S., Ebi, K. L.,
Drewnowski, A., Fukagawa, N. K., and Ziska, L. H.: Carbon dioxide (CO2) levels this century will
alter the protein, micronutrients, and vitamin content of rice grains with potential health
consequences for the poorest rice-dependent countries, Sci Adv, 4,
https://doi.org/10.1126/SCIADV.AAQ1012, 2018.














**Availability Statement**

**All relevant data are included within the manuscript. No additional data, code, or**
**software were used or are available beyond what is presented in the paper.**


**Anton Lokshin and the co-authors**

`



