# Peer review of "Foliar nutrient uptake from dust sustains plant nutrition"

_EGUsphere, 2024_

## Author Comment (AC2)

Dear editor,

We are happy to resubmit our paper, "Foliar nutrient uptake from dust sustains plant nutrition" (egusphere-2024-2531). We were glad to see that the reviewers appreciate the importance of the work we have done. We thank the reviewers for the time and effort invested in reviewing our manuscript. The comments provided were insightful and constructive, contributing significantly to the overall improvement of the paper.

In response to the reviewers' comments, we have made substantial efforts to revise the manuscript. Specifically, we have improved the introduction, clarified the methods, strengthened the discussion section, and toned down the conclusions. Additionally, we have added a new table summarizing all the treatments and updated the previous Table 1 to provide a clearer and more comprehensive presentation of the data.

Our responses to the reviewers are provided below in **bold.** The revised submission includes a version of the paper with a "track changes" on to make it easier for the reviewers and editor to follow the changes we have made in the text (the line and page number in our responses refers to the track changes file only).

Referee #1:

Lokshin et al. show significant foliar nutrient uptake from desert dust and volcanic ash in chickpeas grown under elevated CO2 levels. Using neodymium isotope tracers, they quantified major acquisition of P, Fe, and Ni through leaves rather than roots. This study addresses an important knowledge gap in plant nutrition mechanisms under rising CO2 scenarios.

The experimental design used controlled greenhouse conditions with continuous CO2 fumigation, providing more stable CO2 concentrations compared to traditional Free-Air CO2 Enrichment (FACE) studies, which suffer from fluctuating daytime CO2 levels and nighttime CO2 shutoffs. Furthermore, FACE studies usually have low replication (n = 3) to save on CO2 costs. In contrast, the authors had 12 replicates, if I am not mistaken.

The authors' findings are important for understanding plant nutrition in a high-CO2 world, especially given concerns about declining nutrient content in crops. The research is informative about foliar supplementation as a method to maintain crop nutritional quality under elevated CO2 conditions.

The study has limitations, some of which can be addressed:

1. The shading effect of dust particles on photosynthesis was not fully controlled for, as no inert dust treatment was included to equalize light intensity between groups. While the authors mention this limitation in their discussion, they do not measure the size of potential shading impacts.

   **R: Thank you for your comment. Our study is based on our pervious study that compared the effects of inert silicate dust, which lacks phosphorus, to desert dust with high phosphorus content (Gross et al, 2021, New phytologist). These set of experiments showed no impact of the shading**

**effect caused by the silica on chickpea growth. In another paper by our group (Starr et al., 2023) we examined the impact of foliar application of dust on several tree species, noting dust decreased photosynthesis in some species. To examine the shading effect in our study we assessed the effects of dust and ash on chickpea carbon assimilation (a photosynthesis metric) and found that the positive impacts of dust on biomass and phosphorus content outweighed the negative shading effects on photosynthesis. Based on your comment we have added a paragraph discussing this issue. (P18-19, L)**

1. Statistical analysis is very limited. No statistical power for any of the tests is provided. The number of replicates is not even mentioned in Fig 5 legends for its error bars. The authors should clearly state the number of replicates in the text and all figures that use replicate numbers.

   **R: Thank you for your comment. We have added the statistical analysis and specified the number of replicates in the text (P11,12,21 L).**

3) a few typos and errors easily fixable:

L64 "hidden hunger" is among key words and is relevant for this paper but the authors do not discuss the issue of hidden hunger and CO2 anywhere in the paper. Adding some discussion can be beneficial.

**R: We deleted this key word (P2  L).**

L219 Formula has missing subscripts.

**R: Changed accordingly (P9  L).**

L232 should be X-ray, not X ray.

**R: Changed accordingly (P7  L).**

L281 Table 1 should be foliar-treated not foliar-trated.

**R: The table was completely changed and now it is table 2. We corrected the spelling (P14  L).**

L343 and many other places references repeated twice such as (Hinsinger, 2001)(Hinsinger, 2001) –

**R: Apologies. Technical problems. Corrected accordingly.**

L378 Delete this phrase "Click or tap here to enter text.Click or tap here to enter text.

**R: Apologies. Technical problems. Corrected accordingly.**

L409 messed up font sizes with text in subscripts.

**R: Changed accordingly (P21 L491-495).**

Figures have text boxes that are barely readable due to tiny fonts.

**R: Corrected accordingly (P L).**

Referee #2:

The paper shows that plants can absorb nutrients from deposited dust resulting in increased growth and that the relative benefit can be higher under elevated atmospheric $CO_2$ concentrations. Using Nd isotopes as tracers the authors were able to estimate the contribution of leaf uptake of individual nutrients. The paper thus delivers important and relevant data contributing to our understanding of the importance of dust deposition to plant nutrition, especially under increasing $CO_2$ levels.

However, I think that the paper needs improvement in several aspects, both in general terms and some specific aspects.

The following general aspects need clarification / improvement / discussion:

1. The authors follow a rather teleological approach when interpreting their results. This means they imply that plants induce changes in specific traits on purpose. Examples can be found throughout the paper, especially in the sections with the following headers: "Physiological adaptions toward foliar uptake" (l. 290), "Plant strategies for foliar mineral nutrient uptake" (l.337). I have some doubts about such postulates. First, evolution doesn't follow a plan, and plants are unable to directly, deliberately "adapt" to their environment. They don't have a "strategy". Rather, certain plant traits happen to offer some advantages under specific conditions and are therefore selected during evolution. **Second**, in the present case, the cultivar showing improved nutrient uptake is the product of plant breeding, i.e. human intervention. This excludes that the plants themselves "adapted" or developed "strategies" to benefit from dust deposition. This obviously was the result of coincidence.

   **R: Thank you for the comment. We changed the phrasing throughout the text according to this comment to a "neutral" phrasing. For example, "Plant strategies for foliar mineral nutrient**

uptake" and "Plant strategies for foliar mineral nutrient uptake" changes to "Mechanisms Facilitating Foliar Nutrient Uptake" (P14 L), and "Foliar mineral-nutrient uptake mechanisms" (P18 L) accordingly. In addition, "These changes will derive plants to adapt and search…" changes to "These changes may lead to the selection of plant traits that facilitate alternative nutrient uptake pathways" (P3 L).

2. **A.** The plants were treated with rather high dust doses. The amount of dust was higher than the final dry mass of plants, and the image in Fig. 1 illustrates that dust coverage was extreme. The authors mention that the dust coverage represents the average natural dust deposition in southern Israel during the entire growth period (l. 158/159). It is furthermore not clear whether the entire average deposition per $m^2$ was applied to a single plant or planting density was considered and values were corrected for the area covered by an individual plant. In any case, the relevance of this approach needs more attention and should be further discussed in the paper. **B**. Shading effects should also be discussed in more detail. The authors briefly refer to effects on photosynthesis. Light stress due to high light intensities, which can be expected in this region, may negatively affect plants and dust coverage could thus be beneficial. **C**. In this context: the conditions in the greenhouse (temperature and humidity regimes, light) should be given.

**R: A. We calculated the average foliar area of the chickpea pots, taking into consideration planting density, and correcting values for the area covered by an individual plants. These values were used to determine the total application mass. In total, the average application mass was 3 g per pot, simulating the total dust deposition per square meter over the growing season in southern Israel. This method was based on our previous studies (Gross et al, 2021, Starr et al, 2023, Lokshin et al, 2024). The picture in figure 1 was taken right after application and maybe misleading and with time, the amount of dust retained on the foliage is much lower than 3 grams, and the wet biomass of the plants is significantly higher than their dry weight. Therefore, while our application mass seems high, it was refined in many studies we have conducted to observe the impact of dust on plants without causing significant damage. We added explanation to the text (P6 L). B. We added a paragraph discussing the shading effect, its impact on photosynthesis rates, and the potential for light stress. See also our response to reviewer #1 (P18, L). C: We included the temperature, humidity, and light conditions (P4-5 L).**

3. Some conclusions are too general and not justified by the presented results. In l.440/441 it is stated that "[…] we showed here that dust nutrient uptake via the foliar pathway in chickpea plants plays a major role in their nutrition". This implies that this is the case for all chickpea plants, wherever they grow worldwide. Please keep in mind that you conducted (i) only one experiment (ii) under rather artificial conditions (pot experiment, greenhouse, rather (unnaturally?) high dust doses) and (iii) foliar uptake was only relevant for one specific cultivar. Later (l. 452/453) you conclude that "[…] foliar nutrient uptake from natural dust will play a central role in $eCO_2$ earth […]. Again, I think that is a strong overinterpretation of your results for the above reasons.

**R: Thank you for the comment. We revised the phrasing to: 'We showed that dust nutrient uptake via the foliar pathway can play a significant role in the nutrition of certain chickpea plant types under phosphorus-limited conditions" (P22 L), and "Our findings suggest that foliar uptake from natural dust could be a relevant pathway under future elevated $CO_2$ condition, at least for certain chickpea cultivars" (P22 L).**

4. Please separate the (objective) presentation of results from (subjective) conclusions. Conclusions and interpretations should be restricted to the section "Discussion". In Figs. 1-3 the headings not only explain the results but contain your conclusions: Fig. 1 (l. 266-267) and Fig. 2 (l.275-276): "This implies that […], Fig. 3: "[…] rendering it as more fit to extract nutrients from dust particles".

R: **Thank you for the comment. We separated the conclusions from the headings of Figs. 1-3.**

Specific remarks (from l.1 – 659):

l.34: Root uptake is for a long time no longer considered the "exclusive nutrition pathway" for plants (as also mentioned in the introduction). Please modify sentence accordingly.

**R: Corrected accordingly (P3 L).**

l.82: "has bever been quantified before". I don't agree. A quick search in any database will deliver a whole bunch of respective papers.

**R: As far as we know, this is the first attempt to quantify nutrient flux from dust via the foliage.**

l.82-84: The cited literature on foliar uptake is outdated, and the citations are furthermore missing in the literature list. Please refer to more recent papers (latest edition of Marschner's textbook and references cited therein). You should also mention, at least briefly, the known pathways of foliar uptake and how nutrients bound in solid particles can be absorbed by leaves at all (nutrients in solid form are not available for uptake). In the discussion you explain how acids and sugar may increase availability, but the mechanisms proposed are rather vague and avoid direct explanations.

**R: We have added information regarding foliar uptake in the 'Introduction' section, including citations from Marschner's 4th edition textbook and the references cited therein. Additionally, we revised the discussion to provide a more direct explanation based on Marschner's framework and others (P3 L, P18 L).**

l.86: Please use more precise wording. Accumulation of C always exceeds the that of mineral nutrients.

**R: Changed accordingly (P3 L).**

l.88: "These changes will drive plants to adapt and search for other nutrient uptake". See my general comment #1 above. Plant's IQ is rather low 😊

**R: We changed the phrasing: "These changes may lead to the selection of plant traits that facilitate alternative nutrient uptake pathways" (P3 L).**

l.90: typo. "…of macro…"

**R: Corrected accordingly (P L).**

l.92: "…and for their dependent human and livestock nutrition". Weird sentence, please rephrase.

**R: We deleted this part: "and for their dependent human and livestock nutrition" (P4 L).**

l.93: New paragraph starting with "In this experiment…"

**R: Changed accordingly (P  L).**

1.103: "...a non-responsive genotype": The meaning of this phrase is unclear at this point (later it becomes clear). Responsive to what?

**R: Thank you for this comment. We changed this sentence to: "Additionally, we included a genotype of the wild progenitor C. reticulatum, 'CR934,' which has been observed to exhibit limited responsiveness to foliar application of dust under similar conditions. This allowed us to compare the effects of dust deposition on plant nutrition and leaf properties between a modern chickpea cultivar and its wild counterpart" (P4 L).**

l.114-116: Both sentences are grammatically incomplete, please rephrase. Again: the meaning of responsive is unclear at this stage.

**R: We rephrased the sentences (P4  L).**

l.118: Please specify the climatic conditions in the greenhouses (see comment #2 above).

**R: We added the information regarding climatic conditions: "Temperature was fixed at 25 ± 3°C and relative humidity at 40–50%. Inside the greenhouse the saplings were subjected to natural lighting partially concealed by transparent white walls and roof. Overall, the Photosynthetically Active Radiation (PAR) levels were typical for the southern part of Israel during the months of September to November" (P4,5 L).**

l.122: Typo: liter or litre

**R: Changed accordingly (P  L).**

l.126: Give composition of the nutrient solution and mode of application (intervals, how was it given, from above or below?)

**R: We have added the nutrient solution and mode of application (P5 L).**

l.128: "responsiveness". Again: what does it mean?

**R: We provided additional details regarding the concept of responsiveness (P4 L).**

l.133-137: Please give more details: how did you make sure that all plants uniformly received the same dose (3 g)?

**R: Since dust application was performed manually, we cannot determine the exact amount of dust retained on the leaves with absolute precision. However, we ensured uniformity by applying the same dose (3 g) to all plants, correcting for the average area of the foliage in the pots. It is important to note that due to natural variability, more than 50% of the applied dust likely fell off the foliage during application, but efforts were made to distribute the dust as evenly as possible across all plants (P6 L).**

l.139/140: "…harvested 10 days after the last dust application". You didn't mention so far that dust was applied twice and when the first application took place. This information is given later under the caption "Mineral dust material" (l.145). Please move the information given in l.157-163 to the general description of experimental design (l.133-136).

**R: We moved this information to the general description of experimental design (P6 L).**

l.125-142: The description of treatments is quite confusing. While reading I permanently had to do the math. Why don't you describe your experiment as a 2-factorial design with additional controls plants, n reps each. Think of presenting your design in a table.

**R: We have added a table (Table 1) that summarizes all the different treatments (P5 L).**

l.152: Typo: monthS

**R: Corrected accordingly (P5 L145).**

l.157-163: The information given here is not about the materials themselves (as indicated by the heading) but about the mode of application. Move and merge, see comment above.

**R: This part moved to the 'experimental design' part (P6 L).**

l.166-167: Washing plants without using detergents and/or organic solvents may leave unabsorbed residues on the surface, even if the plain leaf surfaces appear clean. Scanning some random samples by SEM will not safely exclude contamination. Think of particles left in leaf axils or between trichomes. This will lead to a massive overestimation of nutrient uptake. Please consider this when discussing your results.

**R: We appreciate the referee's comment regarding potential contamination from unabsorbed residues on the foliage. However, we adhered strictly to the established washing protocol described in Gross et al. (2021) and Lokshin et al. (2023a), which is designed to effectively remove any residual dust or ash particles from the plant surface. However, we agree that some particles may still be left on the leaves even after our washing procedure (which is a known problem in traditional plant studies also when roots are analyzed). We are not worried about the impact of particle contamination on phosphorus levels, as the P levels in the dust are smaller than in the plant. Thus, simple mass balance calculation shows that remains of few particles on the leaves will not cause a detectable overestimation. In addition, the amount of phosphorus dissolved from the dust using concentrated nitric acid is minimal, further confirming that any residual particles contribute negligibly to the measured phosphorus levels. Unlike P, Fe and Ni may be prone to**

overestimation as their levels in dust are much higher than in plant tissues. We address this concern in the revised text (P20 L**)

Please give explanations (TRU, LN-spec, JNdi, BRC-2)

**R: We have added explanations for TRU, LN-spec, JNdi, BRC-2 (P8 L).**

l.219: missing subscripts

**R: We added the missing subscripts (P L).**

l.281, Table 1: Please give data as means and standard deviation (or similar). Showing individual plant data makes this table rather confusing. Show macro element concentrations in % or mg/g rather than ppm. The value of 713 ppm P is very little for a P-fertilizer. I conclude that the data show the composition of the nutrient solutions, which should be indicated in the table.

**R: Thank you for your remark. We have updated the table to reflect your suggestions, converting the values to averages and standard deviations. Macro-nutrients are now presented in mg/g, while micro-nutrients are given in µg/g. The values observed in the starved chickpea plants are typical for phosphorus-deficient chickpea, consistent with findings in Gross et al. (2021) and Lokshin et al. (2023a). Additionally, the nutritional composition of the nutrient solution and the applied dusts are included in the bottom section of Table 2 (P13 L).**

l.301, Caption Fig.3. Panel b: There are no cases of one or two asterisks, please remove. Panel e: "rendering it more fit to extract nutrients". If consider this an unjustified speculation and not a direct conclusion from the shown results! Trichome density will probably affect retention of dust on leaf surfaces any may be involved in exudation. Both was not in focus of your study and therefore not shown. Furthermore, "extraction" of nutrients again implies a deliberate action of plants (see above). BTW: The only purpose of figures/tables in the section "results" is to transport pure information, not interpretations! Panel f, y-axis: units are missing.

**R: Thank you for this comment. We removed the interpretation sentence from the legend. We added the missing y-axis - the y-axis represents the ratio of the metabolite peak area to the peak area of ribitol, which was added as an internal standard in a fixed amount. This normalization ensures comparability between samples and accounts for variations in wet weight. Ribitol does not influence metabolite secretion but serves as a reference to standardize the data (P15 L).**

L.377/378: remove text.

**R: We deleted the text.**

L.428: "…of the seed". The value is also affected by root uptake of the seedling

**R: We added a sentence explaining that the amount of Nd in the chemical fertilizers was negligible, meaning that all the Nd detected was derived from inheritance (P22 L511).**

l.477: remove double citations throughout the paper (also in the list of references)

**R: Apologies. Technical problems. Corrected accordingly.**

---

## Author Response (AR2)

Dear Editor,

We are pleased to resubmit our manuscript, "Foliar nutrient uptake from dust sustains plant nutrition" (egusphere-2024-2531), following minor revisions. We sincerely appreciate the reviewers' time and effort in evaluating our work. Their insightful and constructive comments have helped us improve the manuscript significantly.

In response to the reviewers' suggestions, we have refined the citations, figures, tables, and bibliography, along with other minor improvements. Additionally, we have incorporated post-hoc p-values in Table 2.

Our detailed responses to the reviewers' comments are provided below in bold. The revised submission includes a tracked-changes version of the manuscript to facilitate the review process. All line and page references in our responses correspond to the tracked-changes document.

Thank you for your time and consideration. We look forward to your feedback.

l. 121: remove double period.

**Corrected accordingly (P4 L121).**

l.134: were applied as (not: in)

**Corrected accordingly (P4 L134).**

l.155/157: remove double caption; CO2 (subscript 2); remove column "Replicates" and give info in the caption (the same in all treatments).

**R: Corrected accordingly (P5 L156).**

l.279: "P starvation did not reduce P concentration". Are you sure? -P concentrations are about 1/3 of the +P controls, which is a substantial difference. BTW: the missing statistics in Table 2 prevents any clear conclusion (see below).

**R: Thank you for pointing out this mistake. The sentence was indeed not clear. We**

**have revised the sentence (P9 L280-L282).**

l.293-301 (Fig.1): Please harmonize the use of upper or lower case for the labelling of panels, both within a Fig. (in Fig.1 upper case in the panels, lower case in the caption) and between the Figs. (mixed in Fig.1 and 2, lower case in Fig.3).

**R: Corrected accordingly (P11 L, P12 L).**

l. 297 (Fig.1): "… was significantly smaller." Smaller than what?

**R: Image was taken immediately after the dust application. (The actual amount of dust remaining on the plant leaves at the end of the experiment was significantly smaller than what is depicted in the picture image). (P12 L316-L317).**

l.301 (Fig.1): The exact meaning of the box plots is not explained. I assume that the central lines within the boxes denote the respective medians, as it is common for this kind of graphs. Furthermore, I am not convinced that the error bars actually denote standard deviations, as stated in l. 301. Standard deviations refer to the symmetric variation of data around arithmetical means. Since there is no symmetry in relation to the central lines, it can be concluded that means are probably not shown in the graphs. Please clarify (also in Fig. 2).

**R: Thank you for pointing this out – this was our mistake. We chose to represent the data using a box and whiskers plot, and here is the explanation: the box and whiskers plots represent the distribution of the data. The central line indicates the median, the edges of the box correspond to the 25th (Q1) and 75th (Q3) percentiles, and the whiskers span from the minimum to the maximum values. Individual data points (n = 5) are overlaid on the plot to illustrate the full distribution. We have updated the text accordingly for Figures 1 and 2 (P12 L316-L318, P14 L351-L353).**

L. 314-317 (Table 2): insert "," before "fertilizers".

**R: Corrected accordingly (P14 L325).**

Results of statistical analyses are missing (post-hoc tests indicating differences within columns). This is important because you refer to those differences in the text of your paper (see my comment above).

**R: Thank you for your feedback. In response to your comment regarding the missing results of statistical analyses, we have now added the p-values from the post-hoc tests to the table, indicating the differences within the treatments (P16 L367).**

l. 319: This header is still too interpretative. Be aware that this is still part of the chapter "results". In the following section you just specify differences between the two cultivars. The interpretation comes later in "discussion".

**R: We changed the header to: "Biochemical properties of chickpea varieties" (P17 L377).**

l. 386: "Marschner, 2022" is wrong. I assume that you refer to chapter 4 of the latest edition of Marschner's textbook, which you cite as "Burkhardt and Eichert" (l.530). The authors given in l. 530 are also wrong, the chapter was written by Eichert and Fernández. Please correct the authors in l. 530 and give "Eichert and Fernández, 2022" as the reference in l.386.

**R: Thank you for this remark. We have made the changes for the citation and bibliography (P21 L444, P27 L642).**

l. 392: Move "(Fig. 3f, fig. S2) to l.391 after "sucrose". The reference to the Figs. belongs to your results, not to the interpretation given in l.391 starting with "likely".

**R: Thank you for this remark. We have made the changes accordingly (P21 L449).**

l.433/434: "These results emphasize […] will be greater…". I think this strong statement should be softened. I agree that your results show that the role of foliar uptake of deposited mineral nutrients may be greater under elevated CO2, but your generalized interpretation goes a bit too far.

**R: Thank you for this remark. The paragraph has been softened accordingly: "These results suggest that the role of foliar uptake of atmospheric nutrients in the mineral nutrition level of plants may increase under $eCO_2$, potentially offsetting some of the nutrient reduction driven by the dilution effect and the downregulation of the root nutrient uptake pathway" (P22 L492-L495).**

l. 438/439: this is your assumption/claim. But how do you justify this statement?

**R: Thank you for your comment. We were very concerned about the potential presence of residual dust particles on the plant surfaces after washing. Thus, we have followed a thorough, three stage-washing protocol that we have established in our lab during previous studies. In addition, we also conducted several experiments to measure whether traces of residual particles that were not washed biased our results.**

**We assume that in the worst-case scenario, remains of 5% of the applied dust particles (from an initial application of 3 grams) on the leaf surfaces after washing. We have mixed a commercial peach leaves powder that is used as a standard for ICP analysis (NIST1547) with 1 to 5% of dust or volcanic ash to investigate the effects of residual particles contamination on nutrient levels. Our results showed that the influence of residual particles became significant only when the percentage of particles on the plant material exceeded 5%, and only for Fe and not for P and Ni.**

**Additionally, we would like to note that in cases where root analysis is performed, there can sometimes be residual soil particles left on the roots despite washing. This is a common occurrence in plant studies, and such residual particles are generally accepted as they do not significantly affect the interpretation of the results.**

**While we believe the information about the typical residual dust amounts is valuable, we think that adding these specific details to the method section is beyond the scope of this paper, which primarily focuses on the main experimental findings.**

l. 551 & 553: Check correct spelling of the names. Fernández, not Ferna´ndez, same for Guzma´n.

**R: Corrected accordingly (P27 L642, L645).**

Please thoroughly check and revise the entire list of references. There are some other inconsistencies, e.g. l. 661 (starting with "V."), l.666 (missing line brake before "Wasserburg"), l.671 (formatting of CO2).

**R: Corrected accordingly (P29 L755, P29 L760).**

My conclusion: After revision this will be a sound and valuable paper. Congratulations to the authors!

---

## Author Response (AR3)

Dear Ms. Vicca,

We sincerely appreciate the reviewer's additional remarks and have carefully addressed them in our revised manuscript. In response, we have:

- Provided detailed responses to the reviewer's comments below in **bold**.

- Added a new table to the Supplementary Materials (Table S2).

- Included additional information regarding the financial support of the study.

- Submitted a revised version of the manuscript with tracked changes and a clean version. All line and page references in our responses correspond to the tracked-changes document.

**Reviewer Comments and Responses**

- Section 3.2 'Elemental analysis of the plants' is very brief and mainly consists of Table 2 without further elaboration. You should either provide a more detailed description of the results (see further comments) or move the elemental concentrations to the Appendix. Since the % change in plant nutrient concentration is shown in Fig. 5, I recommend moving the elemental concentrations to the Appendix.

- Table 2 appears to duplicate some of the biomass information presented in Fig. 1 and 2. Duplication should be avoided. Instead, Table 2 could present the statistics for the data in Fig. 1 and 2, or the statistics could be incorporated into these figures.

R: **Thank you for your feedback regarding Table 2 and Figures 1 and 2. While these figures and the table present related data, they serve complementary purposes. Figures 1 and 2 display individual data points, allowing readers to assess variability and distributions, whereas Table 2 provides summary statistics (means and standard deviations), facilitating a clearer comparison of overall trends. We believe that keeping Table 2 in the main text improves readability and enhances interpretation of the results. However, to minimize redundancy, we have moved the extended dataset to Supplementary Materials (Table S2).**

**Additionally, we have expanded the description of the results in section 3.2 (P12 L354-360).**

p-values were added to Table 2, but without sufficient clarification. It is unclear what treatment effects these p-values compare or even which variable they correspond to (only shoot biomass?). This requires further clarification. All variables and treatment effects should have specified statistics and p-values. The legend of Table 2 should briefly mention the statistical tests used to obtain the p-values.

R: **We have clarified the p-value information in the Table 2 legend, which now reads:**

**"Table 2. Total elemental analysis of plants (Cicer arietinum cv. 'Zehavit'), fertilizers, and dust (ICP-MS analysis). The concentrations of micro- and macronutrients are presented in $\mu g/g$ or $mg/g$, respectively, while plant biomass is shown in grams. The p-values refer to biomass comparisons. For $aCO_2$ treatments,**

**all groups were compared to the -P control grown under aCO₂, and for eCO₂ treatments, all groups were compared to the -P control grown under eCO₂." (P12 L363-366).**

**Additionally, we have added the phrase "P-value Biomass" in Table 2 to further clarify the statistical comparisons.**

In Fig. 5, please change the y-axis label to reflect the variable (rather than just the unit), e.g., "CO₂ effect on plant nutrient concentration (%)".

**R: We have updated the y-axis label accordingly (P20-21 L498-528).**

The table rows contain manuscript page numbers, which should be corrected.

**R: Apologies for the oversight; we have corrected this issue.**

The "Availability Statement" with a signature and text does not align with the manuscript structure. Please remove this information or relocate it to an appropriate section.

**R: We have removed this information from the text (P23 L588-598).**

**In the supplement:**

**Table S2 (P6-7).**

---

## Author Response (AR4)

Dear Ms. Vicca,

First, we sincerely apologize for submitting a version that was not fully finalized. This time, we have carefully addressed all the required corrections with the utmost attention.

As you previously suggested, we removed the elemental analysis of the plants from Table 2 and instead included shoot biomass, root biomass, root-to-shoot ratio, phosphorus (P) concentration, and total P content (calculated as shoot biomass multiplied by P concentration). We also added p-values for each parameter.

Additionally, we have moved Figure 5 to the Results section, as recommended.

We are submitting a revised version of the manuscript, including both a tracked-changes version and a clean version. All line and page references in our responses correspond to the tracked-changes document. Our responses to your comments are provided in **bold**.

- It is still unclear what the p value in Table 2 is for. 'Biomass' is too generic. Should this be shoot biomass? Why are statistics not provided for the other variables in the table? Please provide statistical results for all variables presented in the Table (but see remarks below).

**R: Thank you for your constructive feedback. In response to your comment, we have clarified the p-values in Table 2. The p-values now correspond to comparisons between the control group and the dust and ash treatments under both $aCO_2$ and $eCO_2$ conditions, as well as between the +P and -P treatments under both conditions. We have also provided statistical results for all variables presented in the table, including shoot biomass, root biomass, root-to-shoot ratio, phosphorus concentration, and total P content (P11 L309).**

The text of section 3.2 is named 'Elemental analysis of plants' but only describes biomass responses and does not mention the treatment effects on the elemental concentrations. I previously suggested to remove the elemental analysis from the Table,

but you opted not to. If the elemental analyses are presented in Table 2, you should (1) provide statistical results, (2) describe the treatment effects on the element concentrations and (3) incorporate the findings in the discussion. Moreover, biomass data should be presented in section 3.1, not in section 3.2.

**R: As you suggested previously, we have removed the elemental analysis from Table 2. We also combined Section 3.2 with Section 3.1, now focusing on shoot biomass, root biomass, root/shoot ratio, P concentration, and total P, including their statistical p values, rather than providing full elemental results. Additionally, we have added Section 3.3 to describe Figure 5, which has been moved from the discussion section (P9 L278-283, P14 L379-386, P16 L407-414).**

I had overlooked this before, but similar to Table 2, Figure 5 should (1) report statistical results of the treatment effects, and (2) be incorporated in the Results section. Currently, this figure is only presented in the Discussion, without sufficient statistical support.

**R: Figure 5 has been moved to the Results section as requested (P19 L454-489). Statistical results and p-values are now provided in the figure legend (P19 L488-489).**

Further textual improvements are also needed for the text that is currently in section 3.2. The first 2 sentences merely describe Table 2. Such sentences that merely describe display items ('this table shows that) should be avoided and replaced by a relevant observation (e.g. Shoot biomass was significantly higher for +P than for -P treatments) followed by a reference to the table. Also the further description of the results needs to be improved. On l. 356-357, you write: "The results, based on a post hoc Tukey test, show that +P plants are significantly larger than -P plants." This is not sufficiently clear. Do you mean that shoot biomass was larger? The next sentences have similar issues regarding clarity and need further improvement.

**R: Thank you for your valuable comments. We apologize for the oversight and have revised the text as per your suggestion. We have merged sections 3.2 and 3.1 for improved clarity and flow (P9 L278-283).**

- Please also verify the reporting of the statistics. I noticed that some of the figure legends mention for example 'Asterisks represent statistically significant differences between bars (P<0.05, Tukey test).' This is not a standard way to report statistical results. This should be rephrased: statistical differences are between treatments, not between bars and the Tukey test is only the posthoc test. The ANOVA analysis should be mentioned here.

Put in legend of every figure and make sure you have P value and * in each figure and legend

**R: We have made the necessary corrections as per your suggestion. We rephrased the figure legends to clarify that statistical differences are between treatments and not between bars. Additionally, we now specify that the Tukey test is a post-hoc analysis following ANOVA. We have also ensured that the P values and asterisks are clearly indicated in the legends for each figure (P12 L339-340, P13 L367-368).**

---

## Author Response (AR5)

Dear Dr. Vicca,

Thank you once again for your valuable input. We have made the corrections as you suggested. Below are our point-by-point responses, with the changes highlighted in **bold**.

- Table 2 seems to have 2 legends now, 1 above and 1 below the table. Please correct this and make sure all information is provided in the legend (including an explanation of the p values and underlying statistical test).

**R: We have corrected this by consolidating the two legends into one, ensuring that all relevant information, including the explanation of the p-values and the underlying statistical test, is included (P10  L317-320).**

Section 3.3. contains the following text: "The comparison was conducted as follows: the average value of each nutrient in plants grown under aCO2 was calculated, and then each nutrient in individual chickpea plants grown under eCO2 levels was expressed as a ratio relative to the average under aCO2 conditions (eCO2 plant (each individual plant) / aCO2 plant (average of all the control plants)". As a rule, calculations should not be explained in the methodology section, not in the results. For this simple calculation, however, I suggest to just simplify the text as follows: "To this end, the nutrient content of plants under $eCO_2$ was expressed relative to the average of the $aCO_2$ treatment."

**R: We have revised the text in accordance with your suggestion. The explanation now reads: "To this end, the nutrient content of plants under $eCO_2$ was expressed relative to the average of the $aCO_2$ treatment." (P14 L397-398).**

In general, the manuscript text can be further improved by writing your own results in past tense. In some places, the wording can also be improved. For example, l.280: "The impact of foliar desert and volcanic dust application was reflected by the increase of their biomass and total P content through shoot biomass gain rather than through changes in shoot P concentration" could be improved as follows: "Foliar desert and volcanic dust application increased plant biomass and total P content. This resulted from an increase in shoot biomass rather than through changes in shoot P concentration."

**R: We have corrected the sentence as suggested and have also revised other parts of the manuscript to ensure consistency in using the past tense. (P9 L281-282, L287-L293, L300-L301).**

The data availability statement has been removed in the revised manuscript, but should be added (but without signature - cf. other BG publications).

**R: We have added the data availability statement as requested and included the reference to the supplementary materials in the manuscript. (P21 L602-L608).**